# Implications of Reduced-Complexity Aerosol Thermodynamics on Organic Aerosol Mass Concentration and Composition over North America

Camilo Serrano Damha[1], Kyle Gorkowski[2], and Andreas Zuend[1]

[1]Department of Atmospheric and Oceanic Sciences, McGill University, Montreal, Quebec, Canada
[2]Earth and Environmental Sciences Division, Los Alamos National Laboratory, Los Alamos, New Mexico, USA

**Correspondence:** Camilo Serrano Damha (camilo.serranodamha@mail.mcgill.ca) and Andreas Zuend
(andreas.zuend@mcgill.ca)

**Abstract.** Atmospheric organic aerosol (OA) mass concentrations can be affected by water uptake through its impact on the gas–particle partitioning of semivolatile compounds. Current chemical transport models (CTMs) neglect this process. We have implemented the Binary Activity Thermodynamics model coupled to a volatility basis set partitioning scheme in the GEOS-Chem CTM, providing an efficient reduced-complexity OA model that predicts relative-humidity-dependent mixing and partitioning thermodynamics while limiting the impact on computational efficiency. We provide a quantitative assessment of this water-sensitive OA treatment, focusing on a subdomain over North America. The updated OA scheme predicts a spatiotemporal mean enhancement in surface-level OA mass concentration of 145 % for January 2019 and 76 % for July 2019 compared to GEOS-Chem's most advanced OA scheme. The temporal mean surface-level OA organic mass concentration can increase by up to $\sim 590$ % for January 2019 and $\sim 280$ % for July 2019, with the greatest enhancements occurring over the ocean. The updated OA scheme also quantifies the OA-associated water content. The simulations show how different OA precursors and related OA surrogates contribute and respond to water uptake, including due to changes in temperature and relative humidity over the diurnal cycle in selected winter and summer months. These results are independent of future CTM improvements involving updates to chemical reaction schemes and emission inventories. Our water-sensitive OA scheme allows for a better representation of the seasonal and regional variations of OA mass concentration in CTMs.

## 1 Introduction

Organic aerosol (OA) can be produced in the atmosphere by means of volatile organic functionalization reactions and subsequent gas–particle partitioning. OA can also be directly emitted as primary particulate matter from biogenic and anthropogenic sources. OA accounts for a large mass fraction of ambient fine particulate matter ($PM_{2.5}$) (Murphy et al., 2006; Zhang et al., 2007; Jimenez et al., 2009). Attwood et al. (2014) showed OA often surpasses the contribution of other common $PM_{2.5}$ components, such as sulfate, nitrate, and ammonium species. Burnett et al. (2018) calculated nine million deaths per year worldwide attributable to fine particulate matter due to cardiovascular and respiratory diseases. Health Canada (2021) estimates around 15 thousand premature deaths per year in Canada are associated with exposure to air pollution, in particular, $PM_{2.5}$ and that the

total economic cost due to health impacts is CA\$120 billion annually. In addition to the health impacts, atmospheric aerosol particles, including OA, can affect climate. This is done through their interactions with radiation (i.e., direct aerosol–radiation interactions) and, serving as cloud condensation nuclei (CCN) (i.e., indirect aerosol–cloud–radiation interactions). However, the large spatiotemporal variability of atmospheric aerosols in terms of size ranges, number concentrations, chemical composition, and their microphysical influence on cloud properties represent one the largest uncertainties in human-influenced forcing on the climate system (Forster et al., 2021; Watson-Parris and Smith, 2022). Climate forcing is sensitive to the physicochemical properties of OA, such as its hygroscopicity, which is defined as the ability of OA to take up water vapor from the environment at a given relative humidity (RH). Rastak et al. (2017) studied Earth's radiative budget's sensitivity to OA hygroscopicity and aerosol–climate interactions using climate model simulations. Large-scale 3D atmospheric models usually assign a single, constant hygroscopicity parameter ($\kappa_{org}^{\mathrm{OA}}$) to represent the water affinity of OA. They showed that the choice of $\kappa_{org}^{\mathrm{OA}}$ can have a substantial impact on the estimated average top-of-the-atmosphere radiative fluxes. Since the chemical composition of OA dynamically evolves in the atmosphere over timescales that range from minutes to days (Jimenez et al., 2009), its hygroscopicity also changes considerably in space and time, as reported for laboratory and field OA data (Lathem et al., 2013). Representing the hygroscopicity of OA using a constant parameter can lead to large uncertainties in estimations of the climate impacts of OA.

Water vapor is one of the most important species in the air as an abundant greenhouse gas and due to its impact on PM$_{2.5}$. Particle-phase water content impacts the aerosol chemistry (Volkamer et al., 2007; Surratt et al., 2007; Ervens et al., 2011; Lim et al., 2013; DeCarlo et al., 2018), viscosity (Shiraiwa et al., 2011; Lilek and Zuend, 2022), composition and mass concentration (Zuend et al., 2010; Serrano Damha et al., 2024). Depending on the OA chemical composition and associated hygroscopicity, OA may range from being nearly water-free to a highly dilute aqueous phase in the troposphere. The mixture of organic compounds in the OA is expected to remain typically in a non-crystalline state, ranging from liquid to semi-solid or glassy depending on temperature and composition. The physical state is due to the abundance of ambient water vapor that gets absorbed by the particle or due to the depression of the mixture glass transition temperature (Marcolli et al., 2004). However, the direct and indirect effects of water on the overall OA properties are neglected by most air quality, weather, and climate models due to computational complexity considerations.

Traditionally, modeling OA and other associated air quality metrics on a regional to global scale in the atmosphere is done with a chemical transport model (CTM) such as GEOS-Chem (Bey et al., 2001; Park et al., 2004; Wang et al., 2004; Trivitayanurak et al., 2008; Yu and Luo, 2009; Eastham et al., 2014; Keller et al., 2014; Eastham et al., 2018; Lin et al., 2021; Miller et al., 2024). While not the only source of error in CTMs, the simplification of OA physicochemical processes can lead to large errors in the estimation of OA mass concentration and composition. In the GEOS-Chem three-dimensional model, OA is implemented as an ideally mixed condensed (liquid) phase composed of organic matter only (Chung and Seinfeld, 2002; Pye et al., 2010; Pai et al., 2020), to date without accounting for water uptake. The gas–particle partitioning of SVOCs is the main pathway of OA formation and evolution when an airshed is not influenced by substantial local POA emissions. The partitioning of organic mass between the gas and particle phases is predicted by the GEOS-Chem CTM based on the effective saturation

mass concentration ($C_j^*$) in the gas phase, also referred to as effective volatility, of organic compounds (Pye et al., 2010). This is called a one-dimensional volatility basis set (1D VBS) approach (Donahue et al., 2006).

Hydrophilic organic compounds are made of polar functional groups and have stronger molecular interactions with water (i.e., water solubility) than hydrophobic organic compounds. These compounds, typically containing electronegative atoms such as oxygen, coexist with water in a single water-rich particle phase. Hydrophobic organic compounds, in contrast, either form a separate organic-rich particle phase or partition to the gas phase in the presence of water. The affinity of organic species to water (i.e., their water-seeking properties) thus impacts their mixing behavior in the particle phase. OA water uptake alters the gas–particle partitioning of organic compounds, and thus the equilibrium OA mass concentration (Griffin et al., 2003; Pankow and Chang, 2008; Chang and Pankow, 2010; Gorkowski et al., 2019; Serrano Damha et al., 2024, e.g.). Detailed aerosol thermodynamic models capable of predicting particle-phase nonideal mixing, such as the Aerosol Inorganic–Organic Mixtures Functional groups Activity Coefficients (AIOMFAC) thermodynamic model (Zuend et al., 2008, 2011), are currently not used in large-scale chemistry models like GEOS-Chem. This class of thermodynamic models requires molecule-level chemical structure information, such as the functional groups of organic species present in an aerosol, a type of input that is usually not available. In addition, these models focus on capturing the chemical complexity and associated multicomponent interaction effects, which lead to relatively complex model equations. As a result, they may not meet the computational efficiency requirements of the current generation of CTMs. With the intention of addressing these CTM implementation challenges, an efficient, reduced-complexity OA thermodynamic model called Binary Activity Thermodynamics (BAT) model was developed by Gorkowski et al. (2019) to treat nonideal mixing of organic compounds and water. Serrano Damha et al. (2024) have previously compared a BAT-based VBS (BAT-VBS) framework with two simpler single-hygroscopicity-parameter (Petters and Kreidenweis, 2007) VBS approaches that estimate OA water uptake, such as the ones tested in the Community Multiscale Air Quality Modeling System (CMAQ) (Pye et al., 2017) and used in ISORROPIA-lite (Kakavas et al., 2022). Serrano Damha et al. (2024) have shown that the BAT-VBS model consistently accounts for the variations in $C_j^*$ of organic compounds with RH (aside from temperature), a key feature affecting the predicted OA mass concentration due to a water uptake feedback, which other VBS approaches lack.

In this work, we update the "complex secondary OA with semivolatile primary OA" scheme of the GEOS-Chem model, hereafter referred to as the "dry OA", "default OA" or "standard OA" scheme. Our aim is to improve the GEOS-Chem model predictions of the mass concentration of OA in the atmosphere using the ability of a relatively simple yet thermodynamically consistent model to represent particle-phase nonideality, liquid–liquid equilibrium effects, and the water-sensitive gas–particle partitioning of organic compounds. We implement the BAT-VBS model into GEOS-Chem to capture the feedback effect between OA water uptake and the subsequent re-equilibration of organic mass due to the altered partitioning of semivolatile organic compounds (SVOCs). For gas–particle partitioning predictions, we couple the BAT model with the existing 1D VBS approach of GEOS-Chem. We do not alter the inorganic aerosol treatment of GEOS-Chem, which is solved by the ISORROPIA model (Fountoukis and Nenes, 2007). By design, GEOS-Chem treats OA and inorganic aerosol as if they were completely phase-separated (or externally mixed) by treating their mixing and gas–particle partitioning thermodynamics independently.

**Table 1.** Abbreviations and variables

| Abbreviation or Variable | Definition | Units[a] |
|---|---|---|
| OA | Organic aerosol; referring to organic-rich particle phase only | – |
| LLPS | Liquid–liquid phase separation | – |
| VOC | Volatile organic compound | – |
| IVOC | Intermediate-volatility organic compound | – |
| SVOC | Semivolatile organic compound | – |
| LVOC | Low-volatility organic compound | – |
| ELVOC | Extremely low-volatility organic compound | – |
| $PM_{2.5}$ | Fine particulate matter with a diameter equal to or smaller than 2.5 µm | – |
| AMF | Aerosol mass fraction of organic species | – |
| CTM | Three-dimensional chemical transport model | – |
| 1D VBS | One-dimensional volatility basis set framework | – |
| BAT-VBS | Binary Activity Thermodynamics model coupled to a 1D VBS approach | – |
| MERRA-2 | Second Modern-Era Retrospective analysis for Research and Applications meteorological data | – |
| $O:C_j$ | Elemental oxygen-to-carbon ratio of organic species $j$ | – |
| $H:C_j$ | Elemental hydrogen-to-carbon ratio of organic species $j$ | – |
| $N:C_j$ | Elemental nitrogen-to-carbon ratio of organic species $j$ | – |
| $C_j^*$ | Effective saturation mass concentration of organic species $j$ | $\mu g\,m^{-3}$ |
| $C_j^\circ$ | Pure-component saturation mass concentration of organic species $j$ | $\mu g\,m^{-3}$ |
| $C_j^{gas}$ | Gas-phase mass concentration of organic species $j$ | $\mu g\,m^{-3}$ |
| $C_j^{OA}$ | OA-phase mass concentration of species $j$ | $\mu g\,m^{-3}$ |
| $C_j^{gas+OA}$ | Total mass concentration of organic species $j$ | $\mu g\,m^{-3}$ |
| $C_{org}^{OA}$ | Cumulative OA-phase mass concentration of organic species | $\mu g\,m^{-3}$ |
| $C_w^{OA}$ | OA-phase mass concentration of water | $\mu g\,m^{-3}$ |
| $C_{org+w}^{OA}$ | Total OA-phase mass concentration | $\mu g\,m^{-3}$ |
| $M_j$ | Molar mass of species $j$ | $g\,mol^{-1}$ |
| $\gamma_j$ | Mole-fraction based binary activity coefficient of species $j$ | – |
| $\kappa_{org}^{OA}$ | Hygroscopicity parameter of the OA | – |
| $\xi_j$ | Aerosol mass fraction of organic species $j$ | – |
| $f_{org}$ | Organic mass fraction of a OA compound class | – |
| $f_{SVorg}$ | Cumulative organic mass fraction of semivolatile organic species | – |
| $f_{\Delta org}$ | Organic mass change fraction of a OA compound class | – |
| $f_\beta$ | Mass fraction of organic species in the organic-rich ($\beta$) liquid phase when LLPS happens | – |

[a] The equations and models presented in this work typically use the stated units, which are scaled versions of the preferred units from the International System of Units.

## 2 Methodology

### 2.1 Organic Aerosol Scheme in GEOS-Chem

Atmospheric organic aerosol (OA) was simulated using the GEOS-Chem chemical transport model (CTM) (Bey et al., 2001; Pye et al., 2010; Marais et al., 2016), based on version 14.2.3 (https://github.com/geoschem) released on December 1, 2023 (https://doi.org/10.5281/zenodo.10246546). A nested simulation setup was employed in which the boundary conditions were created using global simulations with a horizontal resolution of $4\,°$ latitude by $5\,°$ longitude. In our nested simulations, a chemical operator duration of 10 minutes and a transport operator of 5 minutes were used. OA mass concentration data were saved every 3 hours. The horizontal resolution of our nested simulations over a North American domain ($-140\,°$ to $-40\,°$ longitude and $10\,°$ to $70\,°$ latitude) was configured at $0.5\,°$ latitude by $0.625\,°$ longitude with 72 vertical levels from 1005.65 hPa (lowest atmospheric level) up to 0.015 hPa (highest atmospheric level) using the second Modern-Era Retrospective analysis for Research and Applications (MERRA-2) meteorological data as distributed with GEOS-Chem. Typically, about eight atmospheric vertical layers constitute the boundary layer (first kilometer from the surface). A buffer zone of three grid cells along each boundary of the North American domain was used. A period of one month was used to spin up GEOS-Chem prior to a targeted simulation time frame.

The full gas-phase chemistry mechanism with the complex secondary OA scheme, considering semivolatile primary OA, was modified as further described in this section and used in all the simulations. The OA species in this work are grouped together into five compound classes corresponding to five main hydrocarbon precursors: terpenes (TSOA/G), isoprene (ISOA/G), light aromatics and intermediate-volatility organic compounds (IVOCs) (ASOA/G), primary semivolatile organic compounds (SVOCs) (POA), and oxidized SVOCs (OPOA) (Pye et al., 2010). The main organic precursor oxidation pathways considered are photooxidation, ozonolysis, and nitrate radical oxidation. As described in Pye et al. (2010), the gas–particle partitioning is parameterized with either a unique one-dimensional volatility basis set (1D VBS) with four effective saturation mass concentration ($C_j^*$) bins, in the case of TSOA and ASOA or an Odum 2-product fit with two $C_j^*$ bins, in the case of POA and OPOA. The reference temperature ($T$) for the $C_j^*$ of TSOA and ASOA species is 298 K. The reference $T$ for the $C_j^*$ of POA and OPOA species is 300 K. Isoprene-derived OA is modeled using an aqueous-phase irreversible reactive uptake scheme by Marais et al. (2016) that replaced the standard 1D VBS approach used in older versions of GEOS-Chem. The probability of each isoprene gas-phase precursor adding mass to the OA is determined using uptake coefficients that are calculated based on aerosol-phase reaction rates and solubilities (Marais et al., 2016).

The default (dry) OA scheme of GEOS-Chem does not consider the impact of the mass concentration of isoprene-derived species on the partitioning of organic compounds simulated by the 1D VBS approach. In our updated (water-sensitive) OA scheme, the presence of ISOA in the OA absorption medium is accounted for by assigning an $C_j^*$ of $0\ \mathrm{\mu g\,m^{-3}}$ to isoprene-derived species predicted by the aqueous-phase irreversible reactive uptake scheme. Their effective volatility is so low that these compounds remain in the particle phase. This modification to the default OA scheme allows us to reconcile the 1D VBS and Odum 2-product approaches of TSOA, ASOA, POA, and OPOA with the irreversible reactive uptake mechanism of ISOA. The higher the mass concentration of pre-existing OA, the higher the aerosol mass fraction (AMF) of organic species (Eq. S3),

even in the absence of OA water. As explained in Sec. 3.2, this addition, however, is not the main source of the predicted enhancement of OA organic mass concentration when applying our updated, water-sensitive OA treatment.

The approximate molar masses ($M_j$) and oxygen-to-carbon ratios (O:C$_j$) of organic compounds needed for our updated BAT-VBS scheme in the GEOS-Chem model were estimated using the molecular corridors approach of Shiraiwa et al. (2014). The molecular corridors allow us to relate pure-component saturation mass concentrations of organics ($C_j^\circ$) to O:C$_j$. The slopes

of the two-dimensional space of $M_j$ vs. $\log_{10} C_j^\circ$ are indicative of the typical increase in molar mass associated with a decrease in volatility. The molar mass vs. volatility slopes limiting the two-dimensional space set the range of O:C$_j$ values expected based on 909 oxidation products considered by Shiraiwa et al. (2014). Among the different compound classes considered, $n$-alkanes have the lowest polarity with an O:C$_j$ of 0, while sugar alcohols are the most polar compounds with an O:C$_j$ of 1. This oxygen content range is used in our work to estimate the O:C of OA species in GEOS-Chem, even though organic

compounds with higher O:C can be found in aerosol samples. In order to make a direct comparison between the BAT-VBS model and the default VBS approach of GEOS-Chem in terms of OA organic mass concentration predictions, we opted to use the same VBS as described in Pye et al. (2010). To use the information given by molecular corridors, $C_j^\circ$ values are assumed to be equal to the (dry-state) effective saturation mass concentrations ($C_j^\circ \approx C_j^*$); the latter are provided by the default OA scheme of the GEOS-Chem model. The default GEOS-Chem model assigns a $M_j$ of $150.0\,\mathrm{g\,mol^{-1}}$ to every TSOA and ASOA

species, which seems unrealistic, especially for oxygenated low-volatility organic compounds (LVOCs) and extremely low-volatility organic compounds (ELVOCs). Likewise, a $M_j$ of $12.0\,\mathrm{g\,mol^{-1}}$ is (incorrectly) assigned to every POA and OPOA species. Those $M_j$ values are irrelevant for solving the equilibrium OA organic mass concentration in the default OA scheme. However, the default values of $M_j$ needed to be corrected for use by the BAT-VBS model in GEOS-Chem, since they are required in the calculation of $C_j^*$ (Eq. 2). The first step to determine the $M_j$ values of organic compounds (volatility bins)

within each of the five precursor types is based on approximating the $M_j$ vs. $\log_{10} C_j^\circ$ slope of each OA precursor. This can be done by estimating the limiting $M_j$ values (i.e., $M_j$ of the most volatile organic compound and $M_j$ of the least volatile organic compound) of each type of OA precursor. The $M_j$ values of the remaining species can then be calculated using the linear equation corresponding to the OA precursor type. For example, in the case of the TSOA compound class, an equation for the linear relationship between $M_j$ and $\log_{10} C_j^\circ$ can be obtained based on the properties of TSOA0 and TSOA3 (Table 2).

Since the $\log_{10} C_j^\circ$ values are provided by the 1D VBS of the GEOS-Chem model, we can then calculate the corresponding $M_j$ values of TSOA1 and TSOA2 using the linear equation of the TSOA compound class. The O:C$_j$ ratios of TSOA0, TSOA1, TSOA2, and TSOA3 can then be approximated using the molecular corridor relationship for every pair of $\log_{10} C_j^\circ$ and $M_j$ values, assuming that O:C$_j$ increases from 0 to 1 between the limits of the red envelope at fixed $\log_{10} C_j^\circ$ in Fig. 1. In the particular case of POA and OPOA, the molecular corridor is instead used to estimate $M_j$ values from every pair of $\log_{10} C_j^\circ$

and O:C$_j$ values given that the GEOS-Chem model provides the global mean organic-mass-to-organic-carbon ratios of POA and OPOA (1.4 and 2.1 for POA and OPOA, respectively), which can be related to O:C$_j$ as follows (Simon and Bhave, 2012):

$$\mathrm{O:C}_j = \left(\frac{12}{15}\right)(\mathrm{OM:OC}_j) - \frac{14}{15}, \tag{1}$$

where OM:OC$_j$ is the organic-mass-to-organic-carbon ratio of species $j$.

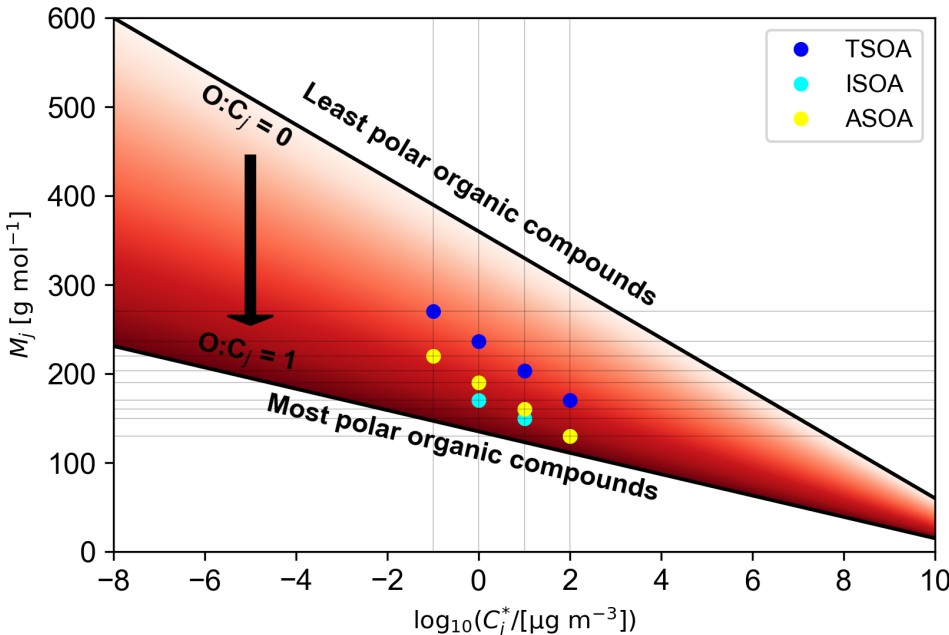

**Figure 1.** Molecular corridors describing the relationship between molar masses, effective saturation mass concentrations at 298 K, and elemental oxygen-to-carbon ratios of OA organic species (adapted from Fig. 4 of Shiraiwa et al. (2014)). The $O:C_j$ ratios of TSOA, ASOA, and ISOA species were estimated using the molar mass vs. volatility space depicted by the red envelope. The least polar organic compounds ($O:C_j = 0$) reside in the upper limit of the envelope. The most polar organic compounds ($O:C_j = 1$) reside in the lower limit of the envelope.

The BAT model assigns one representative functional group to each organic component to represent its water affinity. Of all the options available in terms of functional groups, it was established by Serrano Damha et al. (2024) that using the BAT model with the ketone effective functional group assigned to all organic compounds in binned, VBS-type approaches, produces consistent hydrophilic/hydrophobic behavior in reasonable agreement with the more robust Aerosol Inorganic–Organic Mixtures Functional groups Activity Coefficients (AIOMFAC) thermodynamic model (Zuend et al., 2008, 2011). As such, we decided to set ketone as the effective average functionality that characterizes each organic component (regardless of whether a specific component actually contains ketone functionalities or not).

A schematic of all the OA species considered in GEOS-Chem is shown in Fig. S1. The properties of these species are summarized in Table 2.

## 2.2 Binary Activity Thermodynamics Model

We updated the complex secondary OA scheme of the GEOS-Chem model by coupling the standard 1D VBS described in Sec. 2.1 with an efficient, water-sensitive OA thermodynamic equilibrium model. The Binary Activity Thermodynamics

**Table 2.** Molecular properties of organic compounds used as inputs to run the BAT-VBS model in GEOS-Chem.

| Species[a] | Precursor | $C_j^*$ [µg m$^{-3}$][b,c] | $M_j$ [g mol$^{-1}$] | O:C$_j$ |
|---|---|---|---|---|
| TSOA0 | Terpenes | 0.1 | 270.0 | 0.49 |
| TSOA1 | Terpenes | 1.0 | 236.7 | 0.55 |
| TSOA2 | Terpenes | 10.0 | 203.3 | 0.61 |
| TSOA3 | Terpenes | 100.0 | 170.0 | 0.69 |
| SOAGX | Isoprene | 0.0[d] | 170.0 | 0.84 |
| SOAIE | Isoprene | 0.0[d] | 150.0 | 0.87 |
| LVOCOA | Isoprene | 0.0[d] | 130.0 | 0.90 |
| ASOAN | Light aromatics & IVOC | 0.0 | 220.0 | 0.70 |
| ASOA1 | Light aromatics & IVOC | 1.0 | 190.0 | 0.76 |
| ASOA2 | Light aromatics & IVOC | 10.0 | 160.0 | 0.82 |
| ASOA3 | Light aromatics & IVOC | 100.0 | 130.0 | 0.90 |
| POA1 | Primary SVOCs | 1646.0 | 283.3 | 0.19 |
| POA2 | Primary SVOCs | 20.0 | 232.3 | 0.19 |
| OPOA1 | Oxidized SVOCs | 16.5 | 203.6 | 0.75 |
| OPOA2 | Oxidized SVOCs | 0.2 | 171.9 | 0.75 |

[a] All organic compounds are assumed to have the ketone functionality in terms of the characteristic group type adjustment of BAT when running the BAT-VBS model in GEOS-Chem.

[b] $C_j^*$ values are calculated at a reference $T$ of 298 K for the TSOA and ASOA species. $C_j^*$ values are calculated at a reference $T$ of 300 K for the POA and OPOA species.

[c] The enthalpy of vaporization normalized by the ideal gas constant used in GEOS-Chem is 5000 K.

[d] Isoprene-derived species are assigned an $C_j^*$ of 0 µg m$^{-3}$ in order to reconcile the 1D VBS and Odum 2-product approaches of TSOA, ASOA, POA, and OPOA with the irreversible reactive uptake mechanism of ISOA.

(BAT) model used in this work was developed by Gorkowski et al. (2019). It is a reduced-complexity activity coefficient model that accounts for the nonideal mixing behavior of water and organic compounds in aqueous organic solutions as a function of relative humidity (RH). The BAT model estimates OA water uptake and the (binary) mixture activity coefficients for water and each individual organic compound. In addition, this thermodynamic model can predict the occurrence and extent of liquid–liquid phase separation (LLPS) within aqueous organic mixtures (not accounting for inorganic electrolyte effects in this OA-only case), which can happen when low-polarity organic compounds are present in the OA under high RH conditions. The BAT model and its implementation of LLPS predictions are described in detail elsewhere (Gorkowski et al., 2019; Zuend, 2022b; Serrano Damha et al., 2024). By coupling the BAT model with the 1D VBS of GEOS-Chem (BAT-VBS), we are able to estimate the gas–particle partitioning of organic compounds while accounting for nonideality in the OA, which, as discussed in Serrano Damha et al. (2024), is shown to be important. A main advantage of implementing the BAT-VBS model over more detailed thermodynamic models like the Aerosol Inorganic–Organic Mixtures Functional groups Activity Coefficients (AIOM-

FAC) thermodynamic model (Zuend et al., 2008, 2011) is that it can estimate particle-phase nonideality using only limited information about the OA composition, i.e. information that can be obtained or estimated from the CTM in a straightforward manner, while AIOMFAC would require molecular structure data which are usually unavailable.

185 The typical inputs in each GEOS-Chem grid cell needed to run the coupled BAT-VBS model are: (1) the elemental oxygen-to-carbon ratios $O:C_j$, (2) molar masses $M_j$, (3) effective saturation mass concentrations $C_j^*$ (or, alternatively, pure-component saturation mass concentrations $C_j^\circ$), and (4) the characteristic functional group type of each OA species or the OA overall. The BAT model estimates $C_j^*$ based on a $T$-corrected $C_j^\circ$ and RH. The RH per grid cell is also needed as input, which BAT uses to determine the individual water content contributed to the OA by each organic component. Another key advantage of the
190 BAT model is that, in addition to being a thermodynamically sound method, it is computationally efficient (Serrano Damha et al., 2024), which allows us to directly apply it within CTM simulations. The BAT code was translated into and optimized in modern Fortran (https://github.com/CamiloSerranoDamha/BAT-VBS) based on its original MATLAB implementation that was used for development by Gorkowski et al. (2019).

 In the case of OA consisting of one liquid phase, $C_j^*$, known as the gas–particle partitioning coefficient of each organic
195 component $j$, is given by (Zuend and Seinfeld, 2012; Gorkowski et al., 2019):

$$C_j^* = C_j^\circ \, \gamma_j \, \frac{1}{M_j} \frac{C_{org+w}^{\mathrm{OA}}}{\sum_k \frac{C_k^{\mathrm{OA}}}{M_k}}, \tag{2}$$

where $\gamma_j$ the mole-fraction-based activity coefficient of organic component $j$. $C_{org+w}^{\mathrm{OA}}$ is the total OA mass concentration (Eq. 3), while $C_k^{\mathrm{OA}}$ and $M_k$ are the individual OA mass concentrations and molar masses of OA components, including that of water (i.e., index $k$ covers organic compounds and water). The factor $\frac{C_{org+w}^{\mathrm{OA}}}{\sum_k \frac{C_k^{\mathrm{OA}}}{M_k}}$ represents the mass-concentration-weighted
200 harmonic mean molar mass of the OA. $C_{org+w}^{\mathrm{OA}}$ represents the summation over all components involved (Zuend et al., 2010):

$$C_{org+w}^{\mathrm{OA}} = \sum_k C_k^{\mathrm{OA}} = C_{org}^{\mathrm{OA}} + C_w^{\mathrm{OA}}, \tag{3}$$

where $C_{org}^{\mathrm{OA}}$ and $C_w^{\mathrm{OA}}$ are the cumulative organic and water mass concentrations in the OA phase, respectively (not accounting for the contribution of water due to inorganic salts).

 Our implementation of the BAT-VBS model in GEOS-Chem involved two main steps. First, the molecular properties of
205 organic surrogate species required by the BAT-VBS model had to be estimated, as described in Sec. 2.1. Second, the equations of GEOS-Chem's default OA partitioning scheme (complex secondary OA scheme with semivolatile primary OA) were replaced by the BAT-VBS model. More details regarding this second step can be found in the supplementary information (SI), Sec. S2. The implementation of the BAT-VBS model increases the number of independent variables since the $C_j^*$ values are no longer constant for a given $T$. They are also a function of OA water content (or, indirectly, RH). Water uptake
210 alters the particle-phase mole fractions and mole-fraction-based activity coefficients of organic species $j$, in addition to the mass-concentration-weighted harmonic mean molar mass of the OA (Serrano Damha et al., 2024). As a result, the procedure described through Eqs. (S1)–(S3) needs to be modified as convergence cannot be achieved only by iterating over $C_{org}^{\mathrm{OA}}$ since $C_j^*$ cannot be considered constant. Instead, the BAT-VBS model solves a system of coupled algebraic equations numerically by

**Table 3.** GEOS-Chem simulations performed.

| Simulation Number | OA Scheme | RH conditions | Purpose |
| --- | --- | --- | --- |
| 1 | BAT-VBS model | RH = 0 % | Reference (baseline) simulation that predicts OA mass concentrations assuming dry conditions (forced to run at RH = 0 %). Used as a substitute for GEOS-Chem's standard complex OA scheme with semivolatile primary OA. |
| 2 | BAT-VBS model | RH ≥ 0 % | Simulation that predicts water-sensitive OA mass concentrations using MERRA-2 RH fields as inputs. Used to show the enhancement of OA organic mass concentration and water uptake predicted by the BAT-VBS model in comparison to the baseline simulation. |

iterating over the AMF of organic compounds ($\xi_j$) (Gorkowski et al., 2019). This means that updated values for $C_{org+w}^{OA}$ (Eq. 3)
and $C_j^*$ (Eq. 2) are calculated during each VBS solver iteration step until $\xi_j$ converges, slightly increasing the computational
cost of a nonideal VBS in comparison to the default (dry) VBS approach of GEOS-Chem.

The quantified absolute differences in terms of OA organic mass concentration predictions between the dry (default of
GEOS-Chem) and water-sensitive (updated) OA schemes under dry conditions (RH = 0 %) are small, as demonstrated in
Sec. S3. The absolute and relative differences in OA organic mass concentrations are attributed to the different partitioning
solvers and tolerances used by the two OA partitioning schemes. To estimate more accurately the enhancement of OA organic
mass concentration induced by the water-sensitive OA partitioning scheme at a given RH, we opted to use the predictions of
the water-sensitive OA scheme under dry conditions (RH = 0 %) instead of the default complex secondary OA scheme of
GEOS-Chem as the reference in our calculations. In other words, the OA organic mass concentration enhancement results
shown in this work are obtained from the outputs of two simulations that use the water-sensitive OA scheme (i.e., BAT-VBS
model). The updated version of GEOS-Chem that includes the BAT-VBS model is run twice. These two simulations are listed
in Table 3. In the first run, the BAT-VBS model reads the actual RH values provided by the MERRA-2 meteorological data at
input. In the second run, the BAT-VBS model assumes RH to be equal to 0 %. This latter simulation is used as the reference
(i.e., baseline) in our relative and absolute difference calculations. Since it is a "dry" scheme, it is meant to replicate GEOS-
Chem's default complex secondary OA scheme that accounts for semivolatile primary OA. We decided not to use GEOS-
Chem's default (unmodified) OA scheme directly as there is some minor difference with the water-sensitive OA scheme at RH
= 0 % (Sec. S3). A more accurate estimation of the RH-induced OA organic mass concentration enhancement is obtained by
comparing the outputs of the same OA scheme (i.e., BAT-VBS at RH > 0 % against BAT-VBS at RH = 0 %).

## 3 Results and Discussion

### 3.1 Organic Mass Concentration

The monthly mean OA organic mass concentration predicted at the surface by the water-sensitive OA scheme (BAT-VBS model) is shown in Fig. 2 for January and July 2019. In Figure 2, we decided to focus in a region that is centered on the Southeastern United States due to the area's prominent emissions of natural and anthropogenic organic compounds. Figure S4 illustrates the results from the same simulations but showing the entire North American domain ($10°$N–$70°$N, $140°$W–$40°$W). The surface represents the lowest atmospheric level in GEOS-Chem, which is about 58 m above ground. For January 2019,

the highest monthly mean OA organic mass concentrations are typically found over land, mainly due to TSOA, POA, and OPOA species. For July 2019, the same compound classes are responsible for the hotspot of monthly mean OA organic mass concentrations near the Great Lakes. Moreover, there is an additional contribution from ISOA species to the predicted monthly mean OA organic mass concentrations over land during summer. The absolute and relative differences in mean surface OA organic mass concentrations are calculated using the predictions of the water-sensitive OA scheme at dry conditions (RH =

0 %) as the reference, which is equivalent to the default (dry) OA scheme of GEOS-Chem (Sec. S3). Therefore, positive values of relative and absolute differences throughout the map indicate that the updated water-sensitive scheme (BAT-VBS model) always predicts a higher monthly mean OA organic mass concentration than the dry OA scheme, except at dry conditions (RH = 0 %), at which point the two schemes agree. In January 2019, the absolute difference in monthly mean OA organic mass concentration in the subdomain over North America (Fig. 2b) is mainly due to the RH-sensitive partitioning of the semivolatile

organic surrogate species TSOA2 ($C_j^* = 10\,\mu g\,m^{-3}$), TSOA3 ($C_j^* = 100\,\mu g\,m^{-3}$), and POA2 ($C_j^* = 20\,\mu g\,m^{-3}$), as expressed by their high organic mass change fraction ($f_{\Delta org}$) in Figs. S7 and S10. In July 2019, the absolute difference in monthly mean OA organic mass concentration in the subdomain over North America (Fig. 2e) is also mainly explained by changes in the partitioning of the semivolatile organic species TSOA2, TSOA3, and POA2, as indicated by their high $f_{\Delta org}$ values in Fig. S7. The nonvolatile and intermediate-volatility organic compounds do not contribute to a large extent to the absolute (and relative)

difference in monthly mean OA organic mass concentration predicted between the water-sensitive and dry OA schemes. The water-sensitive partitioning OA scheme predicts a spatiotemporal mean (i.e., averaged over the North American subdomain and month) enhancement in surface organic mass concentrations in the OA (i.e., without considering the OA-associated water) of 145 % for January 2019 and 76 % for July 2019. The monthly mean surface OA organic mass concentrations can increase by up to $\sim$ 590 % for January 2019 and $\sim$ 280 % for July 2019.

We performed a series of shorter simulations to evaluate the sensitivity of our results to reasonable boundary values for the molecular properties of organic species. Figure S12 in Sec. S6 discusses the impact of increasing and decreasing O:C 30 % on the predicted mean OA organic mass concentration enhancement. When normalizing on a per unit percent change of O:C, we found a sensitivity range of 0.36 % to 1.14 % in organic mass concentration enhancement per 1 % change in O:C of the organic compounds.

The fractional contributions of organic PM mass concentration stemming from biogenic and anthropogenic sources predicted by the water-sensitive OA scheme are illustrated in Fig. S5. Biogenic emissions such as isoprene and terpenes are modeled

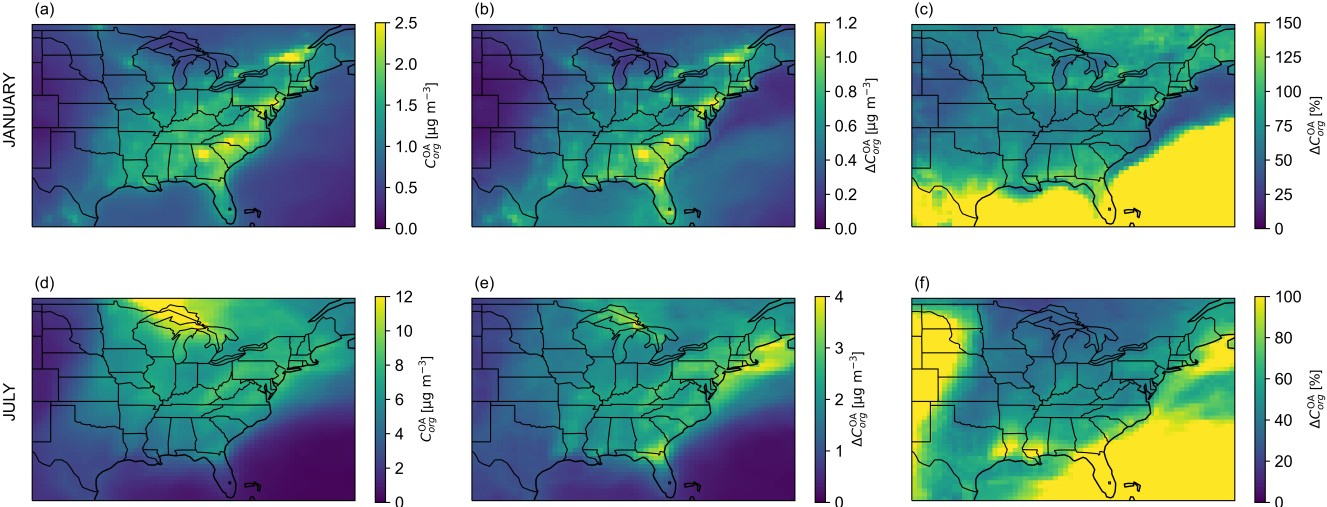

**Figure 2.** January 2019 (top panels) and July 2019 (bottom panels) **(a, d)** monthly mean surface OA organic mass concentration predicted by the introduced water-sensitive OA scheme (BAT-VBS model) at given RH in a region that is centered on the Southeastern United States. Panels **(b)** and **(e)** show the absolute difference in monthly mean surface OA organic mass concentrations. Panels **(c)** and **(f)** show the relative difference in monthly mean surface OA organic mass concentrations. The surface level is the lowest atmospheric level of the GEOS-Chem model. The absolute and relative differences are calculated using the water-sensitive OA scheme (BAT-VBS model) at dry conditions as the reference ($C^{OA}_{org,\mathrm{BAT-VBS}}$ (RH) - $C^{OA}_{org,\mathrm{BAT-VBS}}$ (RH = 0 %)).

using a species-specific emission rate (net in-canopy emission rate at 303 K) that is adjusted according to emission activity factors such as changes in leaf age, air temperature, leaf area index, and light (Guenther et al., 2006; Sakulyanontvittaya et al., 2008; Pye et al., 2010). These factors tend to have higher values in the summer months than in the winter months. The monthly

mean surface OA organic mass concentration for July 2019 is dominated by OA species from both anthropogenic and biogenic precursors, while it is mainly composed of OA species from anthropogenic sources in January 2019 due to the limited rate of biogenic emissions during the winter months. The monthly mean OA organic mass concentration in the North American model domain is greater in July 2019 than in January 2019. The World Health Organization (WHO) recommends an annual mean concentration of particulate matter mass for particles with aerodynamic diameters of less than or equal to 2.5 μm (PM$_{2.5}$) not

exceeding $5\,\mathrm{\mu g\,m^{-3}}$ (World Health Organization, 2021). The January 2019 field of monthly mean organic mass concentration, which is a main component of PM$_{2.5}$, does not exceed that reference air quality target. In July 2019, however, the monthly mean organic mass concentration surpassed that threshold around the Great Lakes, likely due to TSOA, POA, and OPOA emissions and associated impacts from wildfires in western Canada that occurred during that month (Fig. S6a, d, e). Figure S6 shows the individual contribution of each OA compound class to the mean surface OA organic mass concentration, expressed as their

organic mass fraction ($f_{org}$). POA and OPOA tend to be the main source of anthropogenic OA organic mass concentration in

January 2019 over land, while OPOA is the leading anthropogenic OA component in July 2019. OA from TSOA and ISOA constitute the main source of biogenic OA in July 2019 over the continent, while TSOA is the leading biogenic OA component in January 2019.

## 3.2 Water Mass Concentration

Figure 3 shows the monthly mean water mass concentration in the OA at the surface level for January 2019 and July 2019 (panels a, d). Because, to date, the internal mixing of organic species and electrolytes in a single phase is not considered in GEOS-Chem, the water uptake shown is entirely due to the presence of organic species in the aqueous OA. However, note that GEOS-Chem also includes independent predictions of PM mass concentrations of an inorganic aerosol phase and its water content. Three conditions are needed for substantial OA water uptake to occur: (1) the OA must be sufficiently polar

(hygroscopic) to attract water (Fig. 3b, e), (2) the RH must be high enough for sufficient water to be available for equilibrium partitioning to the OA (Fig. 3c, f), and (3) there must be enough organic material in terms of absolute amount that absorbs some water (Fig. 2a, d). For example, in January 2019, the organic mass concentration of OA is highest in the Southeastern United States. That region also corresponds to the highest values of water uptake due to the presence of intermediate-polarity compounds (mean $O:C_{org}$ of 0.5–0.7) and high RH conditions. In July 2019, the hotspot of monthly mean OA water mass

concentration near the Great Lakes corresponds to the high OA organic mass concentration predicted in that region. Once again, the combination of high OA organic mass concentrations of intermediate-polarity compounds (mean $O:C_{org}$ of 0.5–0.7) and high RH conditions explain the predicted high monthly mean OA water mass concentrations near the Great Lakes. The high monthly mean OA water mass concentrations predicted over the northwestern Atlantic Ocean are mainly due to the extremely high average RH in that region for July 2019, in addition to the presence of a high mean OA organic mass concentration and

polarity. The extremely high average RH conditions are the key factor explaining the substantial OA water uptake over the northwestern Atlantic Ocean. In the 99–100 % RH range, water uptake scales exponentially with RH. At those high RH levels, the OA is a dilute aqueous solution mainly composed of water. Since clouds are not resolved in the GEOS-Chem simulation, we decided to set the maximum possible value of RH to 99.5 %, even though the meteorological fields used in GEOS-Chem occasionally suggest higher values.

For January and July 2019, the regions of high monthly mean OA water mass concentrations also correspond to the regions of the highest absolute and relative differences in monthly mean OA organic mass concentrations when comparing the water-sensitive OA scheme with the dry OA scheme (Fig. 2b, e). The reason is that accounting for the mass added by water uptake in the OA promotes the partitioning of organic species from the gas phase to the particle phase (i.e., decreases the AMF of organic compounds (Eq. S3) by two feedback effects. First, the increase in pre-existing OA absorption medium mass

concentration promotes the condensation of semivolatile organic compounds, which in turn will attract more water depending on their hygroscopicity. Second, the decrease in the mass-weighted harmonic mean molar mass and particle-phase nonideality due to the presence of water in the OA decreases the effective volatility of organic species. Consequently, the partitioning of semivolatile organics to the particle phase is enhanced, which triggers additional water uptake in the presence of hygroscopic organic compounds until a new equilibrium state is established. As explained in Serrano Damha et al. (2024), this feedback

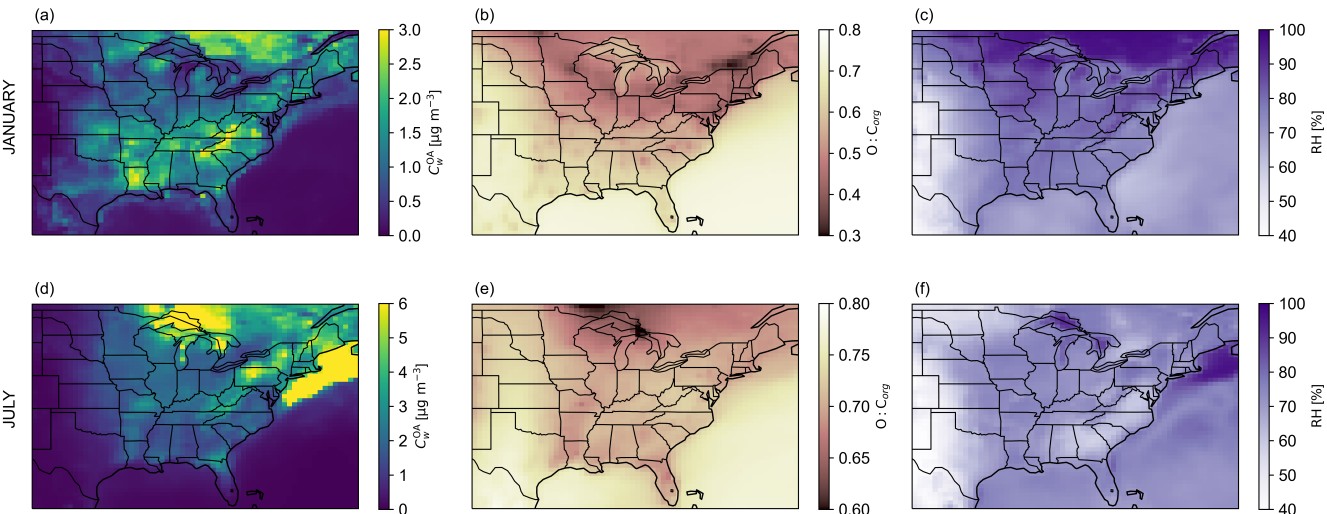

**Figure 3.** January 2019 (top panels) and July 2019 (bottom panels) **(a, d)** monthly mean surface OA-associated water mass concentration predicted by the introduced water-sensitive OA scheme (BAT-VBS model) at given RH in a region that is centered on the Southeastern United States. In panel (d), the color bar was limited to a maximum value of $6\,\mu g\,m^{-3}$, while the highest mean surface OA water mass concentrations reached $\sim 73\,\mu g\,m^{-3}$ over the ocean. Panels (b) and (e) show the monthly mean surface mass-concentration-weighted average O:C of OA. Panels **(c)** and **(f)** show the monthly mean surface RH. The surface level is the lowest atmospheric level of the GEOS-Chem model.

effect is particularly relevant for those semivolatile species that are sufficiently abundant, such that adjustments in their gas–particle partitioning affect the total OA composition. The higher the mass concentration of semivolatile organic species, the more significant this feedback effect becomes. The absolute (and relative) difference in monthly mean organic mass concentration in the OA between the water-sensitive and dry OA schemes (Fig. 2b, e) is mainly due to the decrease in effective volatility of SVOCs with increasing RH, an effect implicitly accounted for by the BAT-VBS model. Capturing the variation of $C_j^*$ with

water content is a feature that both the default (dry) OA scheme of GEOS-Chem as well as single-hygroscopicity-parameter approaches, including the method used by ISORROPIA-Light to estimate "organic" water (Kakavas et al., 2022), are completely lacking.

### 3.3   OA Liquid–Liquid Phase Separation

When multiple organic components are present in the aqueous OA spanning a range of affinities for water uptake (e.g., hy-

drophobic and hydrophilic organics), LLPS may occur as a result of nonideal molecular mixing in the particle phase. The most stable thermodynamic state of the gas–particle system (i.e., the one that has the lowest total Gibbs energy) is then the one associated with the formation of an additional liquid phase, establishing a liquid–liquid equilibrium within the OA. The manifestation of this equilibrium is based on the polarity range of the condensed-phase organic compounds involved, RH, and

$T$ (Zuend et al., 2010; Gorkowski et al., 2019). When LLPS takes place, an organic-rich ($\beta$) liquid phase usually contains the most hydrophobic organic species, while a water-rich ($\alpha$) liquid phase is composed of organic species that have a higher water affinity. It is important to note that the GEOS-Chem CTM assumes by design that electrolytes and organic components are completely phase-separated under all conditions, which is a reasonable assumption when the OA mixture has an average O:C lower than approximately 0.5 (Zuend and Seinfeld, 2012). Here, the consideration of LLPS only concerns the electrolyte-free OA phase. A method for a computationally efficient approximate prediction of the organic phase composition and RH range of LLPS with the BAT model has been introduced in detail elsewhere (Gorkowski et al., 2019; Zuend, 2022a). The expected frequency of OA-specific LLPS events at the surface level for January 2019 and July 2019 is depicted in Fig. 4a, c. In our simulations, each grid cell yields 248 OA organic mass concentration predictions per month. A LLPS event is detected whenever the water-sensitive OA scheme predicts organic mass to exist in both phases $\alpha$ and $\beta$ within the same grid cell.

The January 2019 case resulted in a higher frequency of LLPS events than in July 2019, in particular around Minnesota, Wisconsin, Michigan, Ontario, and Quebec. At these locations, the OA mixture has a low mass-concentration-weighted average O:C ($< 0.5$) (Fig. 3b). This means that the aqueous OA is predominantly made of low-polarity organic compounds that do not tend to mix with water and some higher-polarity organics, in which case an aqueous phase $\alpha$ forms at elevated RH. OA-specific LLPS occurs under high RH conditions ($> 90$ %) only. LLPS happens south of the Great Lakes in January 2019 even though the mean RH can be less than 80 % (Fig. 3c) because some CTM grid cells and time steps have higher RH values during the month. In July 2019, the frequency of OA-specific LLPS events is lower than in January. The map for July 2019 displays a seemingly similar correlation between RH, average O:C, and frequency of OA LLPS events: locations of higher frequency of phase separation events like the Great Lakes correspond to locations of high mean RH ($> 90$ %) (Fig. 3f). This is the case because regions of high mean RH are also associated with regions of moderately hygroscopic OA (mean O:C < 0.7) (Fig. 3e) dominated by POA, OPOA, and TSOA species (Fig. S6f, i, j). The POA species stays in an organic-rich phase separate from a water-rich phase, where the more polar TSOA and OPOA species reside under high RH conditions. The monthly mean mass fraction of organic species in the organic-rich ($\beta$) liquid phase when LLPS happened ($f_\beta$) is shown in Fig. 4b, d. For both months, $f_\beta$ can be as high as 0.8, meaning that when LLPS happens, most of the organic mass is in the liquid phase $\beta$, but a nonnegligible amount of organic mass still resides in the liquid phase $\alpha$.

### 3.4 OA Diurnal Cycle

The high variation of MERRA2-prescribed RH and $T$ fields, in addition to the transport of OA mass concentration between CTM time steps and grid cells, can sometimes complicate the assessment of the updated OA scheme's predictive capabilities. A map of the monthly mean RH field would not necessarily correlate with maps of the monthly mean surface OA organic mass concentration enhancement (with respect to the dry OA scheme), water uptake, and LLPS. An alternative way to analyze the water-sensitive OA scheme behavior is through diurnal cycles and vertical profiles of OA organic mass concentrations that center on a specific geographic location (i.e., CTM grid cell).

Figure 5c shows the monthly mean diurnal cycle of surface OA organic mass concentration for Montreal, Quebec, predicted by the water-sensitive (updated) and the dry OA schemes during July 2019. The dry OA scheme corresponds to the introduced

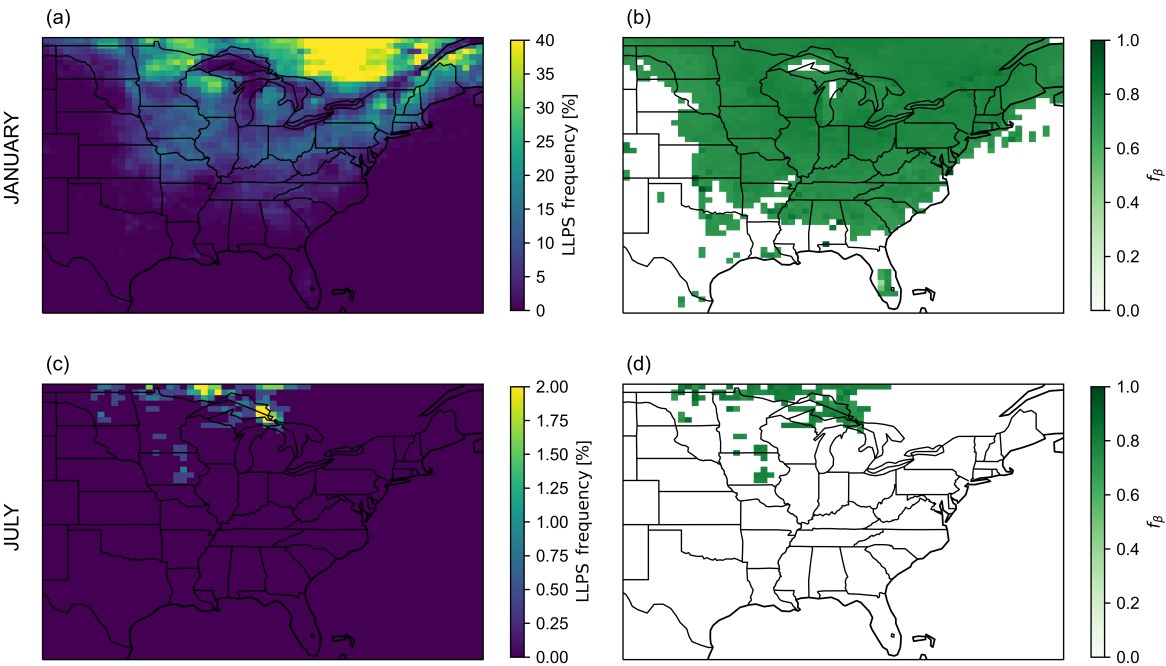

**Figure 4.** Predicted monthly mean surface OA-specific LLPS events for January 2019 and July 2019 in a region that is centered on the Southeastern United States. Panels **(a)** and **(c)** show the frequency of OA LLPS events out of the 248 three-hourly CTM time steps in the month. Panels **(b)** and **(d)** show the monthly mean surface mass fraction of OA species in the organic-rich ($\beta$) liquid phase with respect to the total (liquid phases $\alpha$ and $\beta$) OA organic mass concentration when LLPS happens. Here, LLPS refers to the partitioning of organic species between a less polar organic-rich ($\beta$) liquid phase and a more polar (organic) water-rich ($\alpha$) liquid phase within the OA system. Note that a separate electrolyte-rich aqueous phase exists (not shown/accounted for here). In this study, the water-rich ($\alpha$) liquid phase of the OA system is deprived of inorganic species.

water-sensitive OA scheme (BAT-VBS model) forced to run at RH = 0 %. The solid line and envelope represent the mean value and standard deviation, respectively. The water-sensitive scheme predicts a higher dispersion of OA organic mass concentration data around the mean value than the dry scheme due to its sensitivity to RH.

As expected, the mean RH increases from 5 p.m. to 8 a.m. local time, at which point the maximum value is reached. RH then decreases until the cycle restarts in a similar fashion at 5 p.m. The mean diurnal cycle of atmospheric $T$ follows almost the exact opposite pattern, where $T$ decreases from 5 p.m. to 5 a.m. and reaches its lowest value at 5 a.m. $T$ then increases until the cycle restarts at 5 p.m. The effective volatility of organic compounds typically decreases with decreasing $T$ and increasing RH. The mean absolute (Fig. 5d) and relative (Fig. 5e) differences in OA organic mass concentrations are calculated using the dry OA scheme as the reference. Positive values mean that the water-sensitive OA scheme predicts a higher OA organic mass concentration than the dry OA scheme. The absolute and relative differences in OA organic mass concentrations closely

follow the RH variation over the diurnal cycle: when RH increases (or decreases) with time, so does the difference between the two OA partitioning schemes. While the dry OA scheme completely ignores the equilibrium uptake of water by the particle phase, as demonstrated by the invariability of its predicted OA organic mass concentration to RH in Fig. 5c, the water-sensitive OA scheme captures the impact of water uptake on OA at different moments during the day (i.e., CTM time steps). The mean diurnal cycle predicted by the dry OA scheme over Montreal varies due to $T$ fluctuations and organic mass advection effects. In the case of the dry OA scheme, the OA organic mass concentration peaks at around 5 a.m., which corresponds to the moment of the day when $T$ reaches its lowest value. The Clausius–Clapeyron equation predicts the (dry) $C_j^*$ of organic compounds to decrease as $T$ decreases, which increases the OA organic mass concentration. Even though both OA schemes are exposed to the same $T$ conditions and OA advection effects, the uptake of water and related feedback on the effective volatility of organic compounds that are accounted for by the water-sensitive scheme enhance the predicted OA organic mass concentration over the dry OA scheme. In this particular mean diurnal cycle, the water-sensitive OA organic mass concentration increases by more than 100 % between 2 a.m. and 8 a.m. with respect to the dry OA scheme when RH is around its maximum value.

It is interesting to note that even when the mean RH is greater than 0 %, the two OA partitioning schemes can sometimes match in terms of absolute organic mass concentration predictions in the OA. Figure S13 shows the diurnal cycle of OA organic mass concentration for July 3–4, 2019, over Montreal, Quebec. For example, at 5 p.m. local time, the two OA partitioning schemes predict almost the same OA organic mass concentration. This can happen in three situations. First, this can be the result of having primarily volatile (volatile organic compounds and IVOCs) and practically nonvolatile (low-volatility organic compounds (LVOCs) and extremely low-volatility organic compounds) organic species present in the air mass but a relative lack of SVOCs. In this situation, $C_j^*$ is either relatively high (volatile) or very low (nonvolatile), such that a decrease in effective volatility with increasing RH becomes irrelevant (unlike for SVOCs species). The LVOCs are likely already partitioned to the OA phase regardless of the RH conditions. Second, this can happen when the gas-phase reservoir is depleted of organic species altogether, and there is no organic material left that can partition to the particle phase (Serrano Damha et al., 2024). Third, at dry (low RH) conditions and/or in the presence of low polarity (low O:$C_j$) organic compounds, OA water uptake is modest, limiting the decrease in effective volatility with RH.

### 3.5   OA Vertical Profile

Figure 6c shows the monthly mean vertical profile of particle-phase organic mass concentration predicted by the water-sensitive and the dry OA schemes over Montreal in July 2019, from the surface level to an elevation of 2 km, covering the extent of the planetary boundary layer and the lower free troposphere. The dry OA scheme corresponds to the introduced water-sensitive OA scheme (BAT-VBS model) forced to run at RH = 0 %. The two OA partitioning schemes are used to predict the equilibrium OA organic mass concentration using the conditions available at each vertical level (e.g., the total mass concentration of each organic surrogate compound, RH, and $T$). The water-sensitive scheme predicts a greater spread of OA organic mass concentration data around the mean value than the dry scheme due to its sensitivity to water uptake (and RH). The equilibrium organic mass concentration in the OA tends to decrease as elevation increases, primarily due to the lower mass concentration of OA surrogate compounds at higher elevations away from their near-surface sources. The absolute (Fig. 6d) and relative

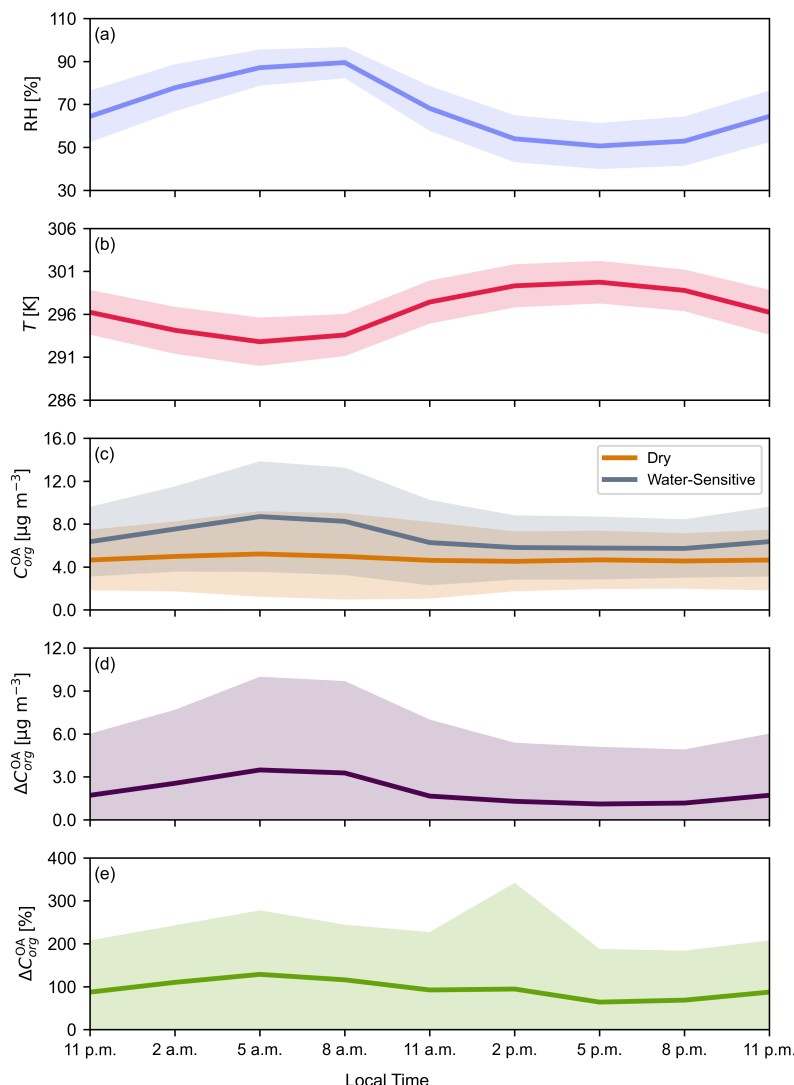

**Figure 5.** Monthly mean time series of three-hourly values of surface RH, $T$, and OA properties predicted by the introduced water-sensitive OA scheme (BAT-VBS model) for Montreal, Canada, during July 2019. The surface level is the lowest atmospheric level of the GEOS-Chem model. The panels show: **(a)** the mean relative humidity, **(b)** the mean temperature, **(c)** the mean OA organic mass concentration predicted by the water-sensitive OA scheme at given RH and the water-sensitive OA scheme at dry conditions, **(d)** the mean absolute difference in OA organic mass concentration, and **(e)** the mean relative difference in OA organic mass concentration. The spread of data around the mean values is depicted with a standard deviation envelope. The absolute and relative differences are calculated using the water-sensitive OA scheme (BAT-VBS model) at dry conditions as the reference ($C^{\text{OA}}_{org,\text{BAT}-\text{VBS}}$ (RH) - $C^{\text{OA}}_{org,\text{BAT}-\text{VBS}}$ (RH = 0 %)).

(Fig. 6e) differences in OA organic mass concentrations predicted by the two OA partitioning schemes are calculated using

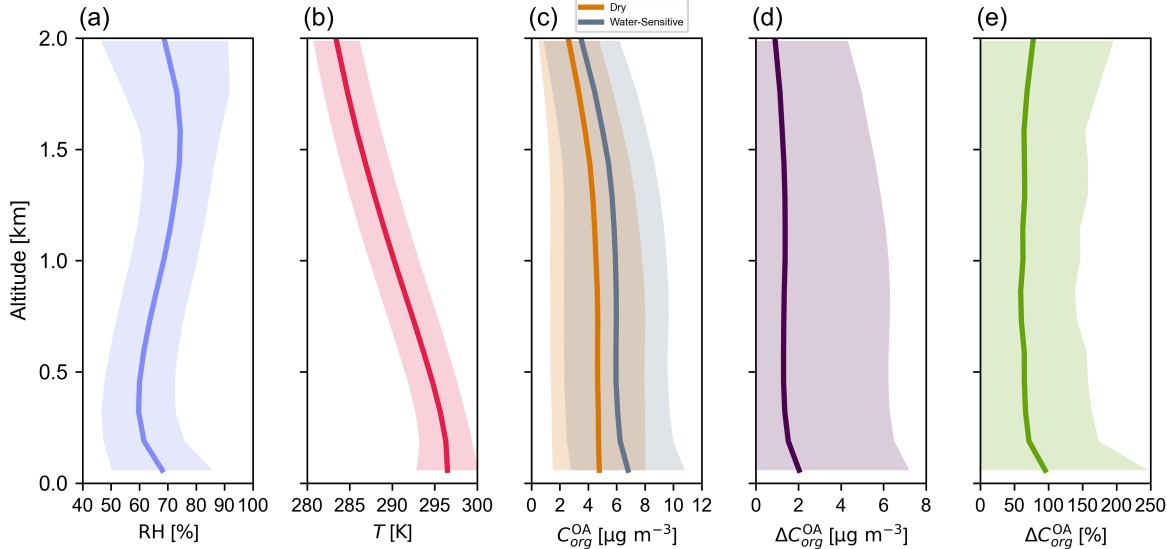

**Figure 6.** Monthly mean vertical profile of surface RH, $T$, and OA properties predicted by the introduced water-sensitive OA scheme (BAT-VBS model) over Montreal, Canada, during July 2019. The panels show: **(a)** the mean relative humidity, **(b)** the mean temperature, **(c)** the mean OA organic mass concentration predicted by the water-sensitive OA scheme at given RH and the water-sensitive OA scheme at dry conditions, **(d)** the mean absolute difference in OA organic mass concentration, and **(e)** the mean relative difference in OA organic mass concentration. The spread of data around the mean values is depicted with a standard deviation envelope. The absolute and relative differences are calculated using the water-sensitive OA scheme (BAT-VBS model) at dry conditions as the reference ($C^{\mathrm{OA}}_{org,\mathrm{BAT-VBS}}$ (RH) - $C^{\mathrm{OA}}_{org,\mathrm{BAT-VBS}}$ (RH = 0 %)).

the dry OA scheme as the reference, which corresponds to the water-sensitive OA scheme at dry conditions. Just as in the case of the mean diurnal cycle (Fig. 5), the vertical profiles of absolute and relative differences in OA organic mass concentrations

display positive values since the mean RH is greater than 0 % in the atmospheric column.

Figure S13 shows the vertical profile of OA organic mass concentration for July 6, 2019, at 5 p.m. local time. In this particular case, even though RH remains above 40 % at an elevation of 2 km, the OA mass concentration predicted by the water-sensitive scheme just slightly surpasses the OA mass concentration predicted by the dry scheme. At that vertical level, the oxidized SVOCs species OPOA2 dominates the organic mass concentration of the OA (Figs. S15b and S16c) in this case. The effective

volatility of OPOA2 species is so low ($C^*_j = 0.2\,\mu\mathrm{g\,m^{-3}}$) that this compounds remains in the particle phase. Consequently, the decrease in volatility with increasing RH becomes irrelevant for the OPOA2 surrogate species and, here, by extension, most of the OA. The water-sensitive and dry OA schemes agree well in terms of OA organic mass concentration prediction at an elevation of 2 km, with the water-sensitive scheme predicting marginally higher OA organic mass concentrations due to the consideration of ISOA species mass and their water uptake in the OA. Unlike the default OA scheme used in GEOS-Chem

that ignores the presence of ISOA species in the absorption medium, ISOA species are assigned an $C^*_j$ of 0 $\mu\mathrm{g\,m^{-3}}$ in our

simulations. This modification to the default OA scheme was done to account for the effect of added ISOA organic mass and its associated water uptake to the OA absorption medium on the partitioning behavior of organic compounds.

The average hygroscopicity of OA is often expressed in large-scale atmospheric models by a single (and constant) hygroscopicity parameter ($\kappa_{org}^{OA}$), which is often set to a value of $\sim 0.1$ (e.g., Rastak et al., 2017; Wang et al., 2019). Because the water-sensitive OA scheme is able to predict water uptake and equilibrium organic mass concentration in the OA, an equivalent effective hygroscopicity parameter can be calculated (Petters and Kreidenweis, 2007). Unlike the traditional simplification adopted by large-scale atmospheric models, our simulations show some modest variations of $\kappa_{org}^{OA}$ with elevation, with $\kappa_{org}^{OA}$ ranging between 0.08 and 0.13 approximately (Fig. S15b). Given the sensitivity of climate models to the value of $\kappa_{org}^{OA}$ assigned to the OA mass fraction (Rastak et al., 2017), capturing the variability of hygroscopicity in space and time, as is possible with the BAT-VBS model, could lead to a better representation of aerosol size changes and related impacts on climate such as aerosol–radiation and cloud–radiation interactions. Due to its high mass fraction (Fig. S16c) and polarity (Table 2), the oxidized SVOCs species OPOA2 dominates the overall hygroscopicity of the OA within the entire 2 km layer.

## 3.6 Comparison between modeled and measured PM$_{2.5}$

There is a need for improvement in several components of OA modeling that affect the predicted OA organic mass concentration, such as emission inventories, number of volatility bins or organic surrogate species considered per OA compound class, OA aging mechanisms, organic aerosol and inorganic aerosol mixing assumptions, etc., making the evaluation of our water-sensitive OA scheme against measurements difficult. Nevertheless, as a validation point for our implementation, we compared modeled and measured ambient fine particulate matter with a diameter equal to or smaller than 2.5 µm (PM$_{2.5}$).

In GEOS-Chem, PM$_{2.5}$ is a combination of mineral dust aerosol, sea salt aerosol, inorganic aerosol, and OA. Figure 7 shows a comparison between modeled and measured daily mean PM$_{2.5}$ mass concentration for July 2019 in monitoring sites located in different states: South Carolina (panels (a) and (b)), Missouri (panels (c) and (d)), Louisiana (panels (e) and (f)), and Pennsylvania (panels (g) and (h)). The top panels (first row) illustrate the cumulative OA organic mass fraction of semivolatile organic species (TSOA2, TSOA3, ASOA2, ASOA3, POA2, and OPOA1) $f_{SVorg}$ as simulated by our water-sensitive OA scheme (BAT-VBS model). For the different monitoring sites, semivolatile organic species represent 10–60 % of the daily mean OA organic mass concentration. The bottom panels (second row) display the modeled and measured daily mean PM$_{2.5}$ mass concentrations corresponding to the same monitoring sites. We considered two GEOS-Chem simulations that differ only in the OA scheme used: (1) GEOS-Chem's default complex secondary OA scheme that accounts for semivolatile POA and (2) our water-sensitive OA scheme based on the BAT-VBS model. The default complex secondary OA scheme that accounts for semivolatile POA refers to the unaltered OA scheme of GEOS-Chem, which is inherently dry. Any discrepancy in the predicted PM$_{2.5}$ between the default and water-sensitive GEOS-Chem simulations (BAT-VBS - Default) is entirely attributed to the different predicted organic OA mass concentrations. The PM$_{2.5}$ time series predicted by the two OA schemes show similar trends, with the water-sensitive treatment always predicting a greater or equal OA organic mass concentration due to its sensitivity to RH. This is due to the fact that winds, temperature ($T$) fields, and emission inventories are the same in both simulations. Both OA schemes are generally consistent with the observed trends of PM$_{2.5}$. One of the main features of the

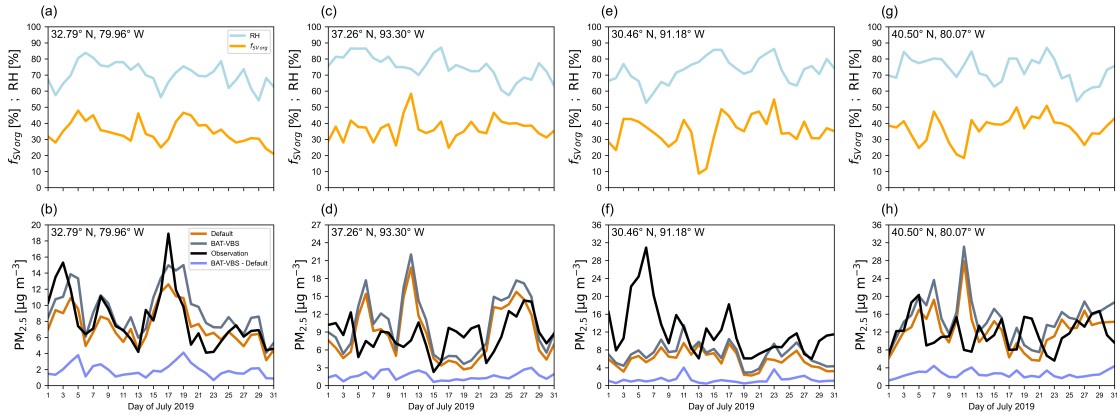

**Figure 7.** Comparison between modeled and measured daily mean PM$_{2.5}$ mass concentration for July 2019 in monitoring sites located in different states. The top panels (first row) show the daily mean RH and daily mean mass fraction of semivolatile organic species in the OA (sum of TSOA2, TSOA3, ASOA2, ASOA3, POA2, and OPOA1). The top panels (first row) show the daily mean RH and daily mean mass fraction of semivolatile organic species in the OA (TSOA2, TSOA3, ASOA2, ASOA3, POA2, and OPOA1). The bottom panels (second row) compare measured daily mean PM$_{2.5}$ to modeled daily mean PM$_{2.5}$. OA is modeled using both GEOS-Chem's default complex secondary OA scheme and our introduced OA scheme based on the BAT-VBS model. Measured PM$_{2.5}$ data comes from US Environmental Protection Agency.

BAT-VBS model is its ability to capture the variation of the effective saturation mass concentration of organic compounds with RH (or OA water content). $C_j^*$ decreases with increasing water content, predicting more organic mass partitioning from the gas phase to the particle phase than the default complex secondary OA scheme of GEOS-Chem. Due to their ability to partition between the gas and particle phases, semivolatile organic compounds are the most affected by the variation of $C_j^*$ with RH (or OA water content). Measured PM$_{2.5}$ data comes from the US Environmental Protection Agency. The ambient air monitors

used often expose the PM$_{2.5}$ sample to low RH and high $T$ conditions in order to dry the aerosol sample. Heating the PM$_{2.5}$ sample often removes both water and a considerable portion of semivolatile organic mass. Given that sampling techniques tend to remove semivolatile organic mass from the OA, a fully adequate comparison between modeled and observed OA (or PM$_{2.5}$) is difficult. Discrepancies between modeled water-sensitive OA and observed OA do not mean that the BAT-VBS is not behaving correctly. In fact, given the limitations of sampling techniques, the BAT-VBS model could provide a more realistic

picture of the actual atmospheric OA mass concentration; however, that is difficult to validate.

### 3.7   Computational Performance

Aside from the impact of the BAT-VBS scheme on OA mass concentrations, the computational cost of the implemented scheme is of practical importance. However, we note that this work was not focused on optimizing computational performance aspects. The water mass concentration due to organic compounds increases the absorptive OA mass while at the same time lowering

the effective volatility of organic compounds. Both effects combined enhance the condensation of organic material to the par-

ticle phase. In GEOS-Chem, when accounting for nonideal gas–particle partitioning, including liquid–liquid phase separation effects, in any grid cell in which RH > 10 %, while using the GEOS-Chem's default complex OA scheme below that RH threshold, the overall wall-clock time of the GEOS-Chem simulation increases by about 50 % with our current implementation. Although the impact of the water-sensitive partitioning scheme on OA mass concentration simulated by the BAT-VBS

model is primarily important for RH > 10 % (see Sec. S9), the water uptake feedback is noticeable for RH > 0 %. We could improve the efficiency of our modified GEOS-Chem simulations by running the BAT-VBS model only under certain atmospheric conditions and reverting back to GEOS-Chem's standard complex secondary OA scheme anywhere else. For example, the BAT-VBS model could be used when RH is above a threshold where the activity coefficients of organic compounds start to deviate significantly from ideality. Another option is to use the BAT-VBS model only within certain atmospheric layers or

levels of interest, such as the surface, the planetary boundary layer, the troposphere, etc. While the work presented in this article is mainly a proof of concept, demonstrating the importance of using a water-sensitive scheme for OA predictions in chemical transport models, further work focusing on improving the implementation of the BAT-VBS model in GEOS-Chem may result in a substantially lower computational penalty. An improvement could involve helping BAT's nonideal VBS solver to converge to a solution faster. The VBS solver, whose objective is to calculate the equilibrium partitioning coefficients of organic

compounds $\xi_j$, accounts for the main computational cost of the BAT-VBS model. By using a memory effect, for example, the BAT-VBS model could reuse $\xi_j$ values calculated in a previous GEOS-Chem time step as initial guesses for the VBS solver. Future work should focus on evaluating various avenues for computational performance improvements.

## 4    Conclusions

To date, the most robust gas–particle partitioning framework implemented as a standard option within GEOS-Chem is the

"complex secondary OA with semivolatile primary OA" scheme. This OA scheme assumes a thermodynamically ideal, water-free organic particulate phase to alleviate the computational cost of OA mass concentration predictions. The gas–particle partitioning of the default OA scheme in GEOS-Chem is parameterized based on (dry) $C_j^*$ bins, which vary only as a function of $T$ according to the Clausius–Clapeyron equation. In this work, we analyzed the extent to which nonideal behavior in the particle phase impacts the gas–particle partitioning of organic compounds as a function of water content (or, indirectly,

RH). The BAT-VBS model, an efficient, reduced-complexity thermodynamic OA model, was implemented into GEOS-Chem to capture variations in $C_j^*$ with RH (aside from $T$), and thus their water-sensitive gas–particle partitioning. We found that the water-sensitive treatment provided by the BAT-VBS model will always predict an enhancement in the OA organic mass concentration compared to the default OA scheme when RH > 0 %.

     We simulated OA mass concentration in a region that is centered on the Southeast of the United States for January and July

2019. We found that the predicted monthly mean surface OA organic mass concentration can increase by up to $\sim 590$ % for January 2019 and $\sim 280$ % for July 2019, with the highest enhancements occurring over the ocean, when using the water-sensitive OA scheme instead of the dry OA scheme. The reason is that at any RH > 0 %, a water-sensitive BAT-VBS-predicted effective volatility of organic compounds (Eq. 2) will always be lower than the corresponding value at dry conditions (RH

= 0 %) that is used by the dry OA scheme (Serrano Damha et al., 2024). Thus, a greater amount of the present semivolatile organic material is expected to partition to the particle phase under realistic atmospheric conditions with the water-sensitive OA scheme, enhancing the OA mass concentration. Coupled to this, the predicted OA water uptake increases the absorption medium (Eq. 3), increasing $\xi_j$ (Eq. S3), the AMF of organic compounds and driving more organic mass from the gas phase to the particle phase. As a result, the hotpots of differences in mean surface OA organic mass concentration predicted by the water-sensitive and dry schemes correspond to the hotpots of predicted OA water uptake. LLPS events in the OA between an organic-rich ($\beta$) liquid phase and a water-rich ($\alpha$) liquid phase were detected by the BAT-VBS model during our nested simulations. These events, which occur as a consequence of nonideal mixing behavior in the particle phase, usually increase the overall OA mass concentration as the partitioning of hydrophobic organic species from the gas phase to the organic-rich particle phase is enhanced when a LLPS happens. In the same manner, hydrophilic organic species have a higher tendency to remain in the water-rich liquid phase as opposed to partitioning to the gas phase when LLPS occurs. In general, for January 2019 and July 2019, LLPS events happened more frequently in grid cells of high RH and low OA mean O:C. In other words, OA-specific LLPS events tend to occur in the lower atmosphere in locations where OA is only moderately polar and the effective gas-phase concentration of water is relatively high. The water-sensitive OA scheme implemented into GEOS-Chem also allows the analysis of vertical profiles and diurnal cycles of OA in the atmosphere. Discrepancies in predicted OA organic mass concentration between the water-sensitive and dry OA schemes are typically detected at CTM time steps and/or grid cells corresponding to RH > 0 %. Exceptions may occur under such RH conditions when OA is mainly composed of low polarity and/or nonvolatile organic compounds. Under those circumstances, the water-sensitive and dry OA schemes can agree in terms of OA mass concentration predictions since the water uptake feedback remains small.

The BAT-VBS model is flexible in terms of the inputs it needs to run, which include the O:C$_j$, $M_j$, $C_j^*$ or $C_j^\circ$, and RH. For physicochemical mixture properties not already available in (and tracked by) GEOS-Chem, estimation methods were implemented to determine those properties in an objective manner for running the BAT-VBS model in GEOS-Chem. In addition, due to its computational efficiency being within the same order of magnitude as that of GEOS-Chem's default OA treatment, the BAT-VBS model is a viable option for improving the physicochemical accuracy of the OA scheme and may help close the gap between GEOS-Chem predictions and field measurements. To our knowledge, no other OA thermodynamic models with those main advantages have been implemented into a major CTM. This work addresses a need and next step in applying bottom-up, process-level models to CTMs. Improving the ability of large-scale atmospheric models to capture the variability of OA properties is needed to more accurately assess the health impacts of OA, improve air quality predictions, and reduce the large uncertainties in aerosol radiative forcing of the climate system.

*Code availability.* The modern Fortran source code of the BAT-VBS model (v1.0.0) is preserved in Serrano Damha (2023b) and developed openly in Serrano Damha (2023a) under a GNU GPL license v3.0.

*Data availability.* The GEOS-Chem output data used to produce the figures of this manuscript are freely available in Serrano Damha (2024). The MERRA2 data used in this work have been provided by the Global Modeling and Assimilation Office (GMAO) at NASA Goddard Space Flight Center.

*Author contributions.* CSD developed the water-sensitive organic aerosol partitioning scheme of the GEOS-Chem chemical transport model and performed the nested GEOS-Chem simulations. CSD, AZ and KG discussed model results and implications. CSD prepared the manuscript, with contributions from KG and AZ. All co-authors were involved in formulating the research goals and aims. AZ conceived the project, was responsible for the supervision and secured funding for this project.

*Competing interests.* The authors declare that they have no conflict of interest.

*Acknowledgements.* This project was undertaken with financial support from the government of Canada through the Federal Department of Environment and Climate Change (grant no. GCXE20S049), the Natural Sciences and Engineering Research Council of Canada (NSERC grant no. RGPIN-2021-02688) and the Canada Foundation for Innovation (CFI John R. Evans Leaders Fund 33530). Additionally, this research was enabled in part by support provided by Calcul Québec (https://www.calculquebec.ca) and the Digital Research Alliance of Canada (https://alliancecan.ca). KG was supported primarily by the U.S. Department of Energy's Atmospheric System Research (project F3LB, PI: Allison C. Aiken), an Office of Science Biological and Environmental Research program; Los Alamos National Laboratory is operated for the DOE by Triad National Security, LLC under contract [Contract No. 89233218CNA000001]; LA-UR-24-29120.

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
