# Peer review of "Implications of Reduced-Complexity Aerosol Thermodynamics on Organic Aerosol Mass Concentration and Composition over North America"

_EGUsphere, 2024_

## Author Comment (AC1)

**Response to Reviewers**

**Camilo Serrano Damha**

The authors would like to thank both reviewers for their careful evaluation of our manuscript and their constructive comments. We address the comments in the following and indicate related changes made to the revised version of the manuscript.

**1 Reviewer 1**

1. First, after multiple readings I am still not completely clear on the details of the "RH = 0" dry case comparison. As described in the supplementary information, most figures compare the modified simulation to itself, using RH = 0 conditions as a baseline: "in this work, we compared the water-sensitive OA scheme at RH > 0 % with the water-sensitive OA scheme at RH = 0 %". Does this mean that the updated version of GEOS-Chem is run again, but with RH in all BAT calculations pinned to 0? Some other model modification? Or is this a filtering process for comparison of actual modeled conditions at low vs. high humidity? Details here should be clarified, considering their importance for the interpretation of all difference plots. If there are multiple simulations being performed (base, modified, modified with RH=0, etc) they should be clearly listed, named, and described in a main manuscript table..

   **Authors' response:** We performed two main types of simulations in this work, both of which use the water-sensitive OA scheme (i.e., BAT-VBS model). The updated version of GEOS-Chem that includes the BAT-VBS model is indeed run twice.

   In the first type of GEOS-Chem simulation (simulation 1), the BAT-VBS model reads the actual RH values provided by the MERRA-2 meteorological data in every grid cell at every aerosol/chemistry time step during the simulation. Simulation 1 aims to highlight the implications of using a water-sensitive OA scheme on predicted OA organic mass concentration.

   In the second type of GEOS-Chem simulation (simulation 2), the BAT-VBS model is also used but assumes the RH to be equal to 0 %. Simulation 2 is used as the dry reference (i.e., baseline) in our relative and absolute difference calculations. Since it is run as a "dry" scheme, it is meant to replicate GEOS-Chem's default complex secondary OA scheme that accounts for semivolatile primary OA.

   We decided not to use GEOS-Chem's default complex secondary OA scheme directly as the baseline simulation since there is some minor difference with the water-sensitive OA scheme at RH = 0 %. This is explained in Sec. S3. A more accurate estimation of the RH-induced OA organic mass concentration enhancement is obtained by comparing the outputs of the same OA scheme (i.e., BAT-VBS at RH ≥ 0 % against BAT-VBS at RH = 0 %).

   GEOS-Chem's default (unmodified) complex secondary OA scheme is used to produce two figures: (1) a first figure to compare it to the BAT-VBS model at dry conditions (forcing RH to be equal to zero) in terms of OA organic mass concentration in Sec. S3 of the supplemental information and (2) a second figure to compare it to observations (measurements) in terms of $PM_{2.5}$ in Sec. 3.6 of the main manuscript. This has now been clearly indicated in Secs. S3 and 3.6.

   **Manuscript modifications:** We described the different simulations performed more clearly in Sec. 2.2 of the main manuscript. Additionally, the simulations performed in this work are now listed in Table 3 of Sec. 2.2.

2. It appears to me that one major uncertainty in this new implementation of BAT is that of assigned O:C ratios for the binned and simplified GEOS-Chem species. I understand the necessity of the approach taken here (as described in SI), but I have to wonder at the sensitivity of final results to variability in these assumed properties. A series of sensitivity tests using boundary values for reasonable O:C ranges would help to quantify how sensitive model results actually are to this uncertainty and simplification.

**Authors' response:** We agree. We performed a series of shorter simulations to evaluate the sensitivity of our results to reasonable boundary values for the molecular properties of organic species. Figure S12 shows that comparison, and we discuss the impact of increasing and decreasing O:C by 30 % on the predicted mean OA organic mass concentration enhancement in Sec. S6 of the supplemental information. To calculate the OA organic mass concentration enhancement, we subtracted the predictions of the water-sensitive OA scheme at dry conditions (forced to run at RH = 0 %), which is our baseline simulation, from the predictions of the water-sensitive OA scheme (RH ≥ 0 %) that uses different boundary values for the O:C of organic compounds. When normalizing on a per unit percent change of O:C, we found a sensitivity range of 0.36 % to 1.14 % in organic mass concentration enhancement per 1 % change in O:C of the organic compounds.

**Manuscript modifications:** Section S6 and Fig. S12 were added to the supplementary information to quantify and discuss sensitivity of the model to O:C of organic compounds. Section S6 is referred to in Sec. 3.1 of the main manuscript.

3. While the modeling results on their own are very interesting and helpful, there is a notable absence in this manuscript of comparison to observations. Of course modeled OA in general has many areas needing improvement, making the comparison a tricky one, but there is value in noting how these mechanism improvements and changes translate to real world comparisons. I think some form of comparison with meaningful observations is a reasonable expectation here, no matter how good or bad the impact on agreement may be.

**Authors' response:** A comparison between modeled OA and observed OA can be complicated because many sampling techniques involve exposing the aerosol sample to low RH and high $T$ conditions to dry it. Heating the aerosol sample often removes both water and a considerable portion of semivolatile organic mass. One of the main features of the BAT-VBS model is its ability to capture the variation of the effective saturation mass concentration of organic compounds ($C_j^*$) with RH (or OA water content). $C_j^*$ decreases with increasing water content, predicting more organic mass partitioning from the gas phase to the particle phase than the standard complex secondary OA scheme of GEOS-Chem. Due to their ability to partition between the gas and particle phases, semivolatile organic compounds are the most affected by the variation of $C_j^*$ with RH (or OA water content). While sampling techniques tend to remove semivolatile organic mass from the OA, the GEOS-Chem model could predict significant OA organic mass in the semivolatile bins, which will make the comparison and interpretation between modeled and observed OA less straightforward. We initially did not want to focus on comparisons with observations in our work because, as pointed out by the reviewer, modeled OA has many areas needing improvement, such as emissions, number of volatility bins or organic surrogate species, OA aging mechanism, internally mixed organic and inorganic aerosols assumption, etc. Discrepancies between modeled water-sensitive OA and observed OA (or observed $PM_{2.5}$) do not mean that the BAT-VBS is not behaving correctly. In fact, given the limitations of sampling techniques, the BAT-VBS model could provide a more realistic picture of the actual atmospheric OA mass concentration. Within the current version of GEOS-Chem, the BAT-VBS model and GEOS-Chem's default complex secondary OA scheme will follow the same trends of OA mass concentrations because both models are subjected to the same winds, emissions, $T$, etc., with the BAT-VBS model always predicting a greater or equal OA organic mass concentration due to its sensitivity to RH. Both OA schemes are roughly in agreement with the observed trends of $PM_{2.5}$. This is shown now in Fig. 7 and discussed in Sec. 3.6.

**Manuscript modifications:** We added Sec. 3.6 in the main manuscript to compare modeled and observed $PM_{2.5}$.

4. It's not clear to me why the model domain cuts off the west coast of the United States. This should be addressed or more clearly explained.

**Authors' response:** Even though we run GEOS-Chem simulations for the entire North American domain (10°N–70°N, 140°W–40°W), we wanted to concentrate our model–model comparisons plots in a region that is centered on the Southeastern United States due to the area's prominent emissions of natural and anthropogenic organic compounds. Our work demonstrates the importance of the feedback effect between water uptake, changing particle properties, and the subsequent uptake of additional semivolatile organic compounds from the gas phase to the particle phase. Even though the domain shown focuses on the east coast of the United States, the water-sensitive scheme effect on the gas–particle partitioning of organic species is important in any area that has some organic mass concentration in the gas phase, especially in the semivolatile bins of the VBS, and RH > 10 %.

**Manuscript modifications:** We made a version of Fig. 2 that shows the entire North American domain (10°N–70°N, 140°W–40°W) and added it to Sec. S4 of the supplementary information.

5. The supplementary information seems unusually extensive to me, and includes some figures and descriptions that I think are crucial to the overall manuscript narrative. I recommend looking over this content carefully and considering whether or not some of it should be moved to the main manuscript.

**Authors' response:** We agree. After multiple readings, we realized that the section describing the derivation of the molecular properties of organic species is crucial to the narrative.

**Manuscript modifications:** We decided to move part of Sec. S2 (Implementation of the BAT-VBS model in GEOS-Chem) of the supplementary information to Sec. 2.1 (Organic Aerosol Scheme in GEOS-Chem) of the main manuscript to expand on the derivation of the organic structure information of OA species.

6. A 50 % increase in wall clock run time is pretty massive, and probably not acceptable for most modelers. The manuscript text mentions possibilities for computational efficiency improvements. Is there any sense of how much these might mitigate the computational cost of incorporating these improvements?

**Authors' response:**

The introduced water-sensitive OA scheme's main computational cost is associated with its nonideal VBS solver. Unlike the RH-independent volatility bins used by the standard complex OA scheme of GEOS-Chem, the water-sensitive effective saturation mass concentrations ($C_j^*$) calculated by the BAT-VBS model depend on the activity coefficients of organic species, and the particle-phase mole fraction of organic species and water (Eq. 2). In terms of solving for the aerosol mass fraction of organic species ($\xi_j$) (Eq. S3), the BAT-VBS model does not simply iterate over the cumulative OA organic mass concentration ($C_{org}^{OA}$) because $C_j^*$ values are not constant, they depend on the OA water content. Instead, the BAT-VBS model must iterate over partitioning coefficients $\xi_j$. The implemented water-sensitive OA scheme uses deep learning neural networks to provide reasonable initial guesses for $\xi_j$ values, thereby helping the nonideal VBS solver find an equilibrium state. However, we believe convergence to an equilibrium state could be sped up by reusing the equilibrium $\xi_j$ values calculated in the previous GEOS-Chem time step as initial guesses for the nonideal VBS solver. Assuming advection effects are small from one time step to the other in a GEOS-Chem grid cell, the aerosol partitioning coefficients of organic species from a previous time step may be closer to the VBS solution than initial guesses provided by neural networks, ultimately speeding up the convergence to a new equilibrium state.

A small fraction of the computational cost of the BAT-VBS model is associated with calculating the mole-fraction-based activity coefficients of organic species. We do not believe that modifying this step would

have a noticeable impact on the current computational cost of the BAT-VBS model in GEOS-Chem. Since BAT-predicted activity coefficients are calculated at each time step in each grid cell of the GEOS-Chem model, we could try reducing the computational burden of this process by using look-up tables of precalculated activity coefficients. At different RH values, the activity coefficient of a given organic species can be predicted by BAT using organic structure information, including O:C and molar mass. Activity coefficient data could then be defined and read from a designated GEOS-Chem Fortran module.

Some quick changes that would improve the efficiency of the updated GEOS-Chem model involve running the BAT-VBS model only under certain atmospheric conditions and reverting back to GEOS-Chem's standard complex secondary OA scheme anywhere else. For example, the BAT-VBS model could be used when RH is above a threshold where the activity coefficients of organic compounds start to deviate significantly from ideality. Another option is to use the BAT-VBS model only within certain atmospheric layers or levels of interest, such as the surface, the planetary boundary layer, the troposphere, etc.

One would have to actually implement the modifications discussed above to assess how much they can mitigate the computation cost of the water-sensitive OA scheme in GEOS-Chem. While the work presented in this article is mainly a proof of concept, demonstrating the importance of using a water-sensitive scheme for OA predictions in chemical transport models, future work will be needed to focus on improving the computational efficiency of the BAT-VBS model in GEOS-Chem.

**Manuscript modifications:** We updated Sec. 3.7 of the manuscript to mention possible ways to improve the efficiency of our updated OA scheme.

**2 Reviewer 2**

1. Equation 1: How is the molar mass of water included? Line 157 indicates it may be in the sum over k? It seems like the text may need to be better synced with the equation.

   **Authors' response:** Below Eq. (2) of the main manuscript, we stated that $C_k^{\mathrm{OA}}$ and $M_k$ are the individual OA mass concentrations and molar masses of OA components (including that of water). The summation index $k$ indeed covers the individual organic compounds and water.

   **Manuscript modifications:** We updated the sentence below Eq. (2) of the main manuscript to specify that the sum over index $k$ includes individual organic compounds and water.

2. Figure 1e—the outflow from NYC and northeast US seems notably enhanced. What drives the enhancement?

   **Authors' response:** The enhancement is driven by the high relative humidity conditions (mean RH $\geq$ 60 % in Fig. 3f) and high polarity (mean O:C $\geq$ 0.70 in Fig. 3e) of the organic species in the particle phase, which triggers a significant amount of water uptake in the OA (Fig. 3d) over NYC and northeast US. The high water content significantly lowers the effective saturation mass concentration of organic species, driving more organic mass to the particle phase and explaining the enhancement of predicted OA organic mass concentration with respect to a dry treatment (Fig. 2e). Figure S7 shows the monthly mean surface contribution of OA species from terpenes (TSOA) to the absolute difference in OA organic mass concentration, expressed as the organic mass concentration change fraction $f_{\Delta org}$. According to panels (g) and (h), TSOA2 ($C_j^* = 10\ \mu\mathrm{g\,m^{-3}}$) and TSOA3 ($C_j^* = 100\ \mu\mathrm{g\,m^{-3}}$) organic species were the main contributors to the enhancement of predicted OA organic mass concentration with respect to a dry treatment. A water-sensitive OA scheme mainly affects the gas–particle partitioning of semivolatile organic species, as explained in Serrano Damha et al., 2024.

3. Line 277 indicates the inorganic electrolytes are phase separated for O:C < 0.5. Figure 2 shows O:C is generally high. Does this suggest the electrolytes should be mixed with OA?

   **Authors' response:** Molecular interactions between inorganic and organic species might become more relevant in all aerosol phases when the mean O:C of organic species is greater than $\sim$ 0.6. One would need a more complex treatment than the current version of the BAT model to predict whether the atmospheric aerosol will be a liquid one-phase particle or whether significant divergence from ideal mixing between organic and inorganic species will trigger liquid–liquid phase separation. We believe that GEOS-Chem's phase-separated assumption of the atmospheric aerosol into internally mixed organic and inorganic aerosols might be reasonable under certain conditions. For example, more robust thermodynamic predictions that focus on particular systems of secondary organic aerosol, water and electrolyte (Zuend and Seinfeld, 2012) agree that liquid–liquid phase-separation is more likely at low RH and low mean O:C of organic species.

4. Figure 4: Consider shifting the time axis to run midnight to midnight. It took a minute to realize it was another time. Where are 6, 7pm?)

   **Authors' response:** The time resolution of our GEOS-Chem outputs is three-hourly, meaning that every simulation day consists of eight data points only: 2 a.m., 5 a.m., 8 a.m., 11 a.m., 2 p.m., 5 p.m., 8 p.m., and 11 p.m.). We do not have the data for 6 p.m. and 7 p.m. due to the output time resolution used.

   **Manuscript modifications:** We shifted the time axis in Figs. 5 and S13 to show the daily cycle more clearly. The axis now starts and ends at 11 p.m.

5. Consider a rename of section 3.2 as model performance is often used in reference to how predictions perform relative to observations.

   **Authors' response:** We agree.

   **Manuscript modifications:** We renamed Sec. 3.7 of the revised manuscript as Computational Performance.

**References**

Serrano Damha, C., Cummings, B. E., Schervish, M., Shiraiwa, M., Waring, M. S., & Zuend, A. (2024). Capturing the relative-humidity-sensitive gas–particle partitioning of organic aerosols in a 2d volatility basis set. *Geophysical Research Letters*, *51*(3), e2023GL106095. https://doi.org/https://doi.org/10.1029/2023GL106095

Zuend, A., & Seinfeld, J. H. (2012). Modeling the gas-particle partitioning of secondary organic aerosol: The importance of liquid-liquid phase separation. *Atmospheric Chemistry and Physics*, *12*(9), 3857–3882. https://doi.org/10.5194/acp-12-3857-2012

---

## Author Response (AR2)

**Response to Reviewer Comments**

Camilo Serrano Damha

The authors would like to thank both reviewers for their careful re-evaluation of our updated manuscript and for their constructive comments. We address the comments in the following and indicate related changes made to the revised version of the manuscript.

**1 Reviewer 2**

1. Can the simulation names be consistent with table 1 with the addition of a "default"? Why are the default and BAT default so different?

   **Authors' response:** In the legend of Figure 7, the "Default" simulation refers to GEOS-Chem's default (unmodified) complex OA scheme. The "BAT-VBS" simulation refers to GEOS-Chem's water-sensitive (introduced) simulation that uses the OA scheme based on the BAT-VBS model. The notation "BAT-VBS - Default" corresponds to the difference between the two simulations above in terms of predicted $PM_{2.5}$ mass concentrations. The BAT-VBS simulation will always predict more OA mass concentrations than GEOS-Chem's default complex OA scheme since RH is typically greater than 0 % in the atmosphere, leading to higher daily mean $PM_{2.5}$ mass concentrations.

   **Manuscript modifications:** We updated the legend, caption, and accompanying text of Figure 7 to explain more clearly what "Default" simulation means and clarify that "BAT-VBS - Default" corresponds to a difference among the predicted $PM_{2.5}$ mass concentration data.

2. Add descriptive location names to fig 7 panels. Are the panels a specific point in latitude, longitude or an entire state? How were sites selected?

   **Authors' response:** The daily mean $PM_{2.5}$ mass concentrations displayed in Figure 7 correspond to specific geographical locations (i.e., the monitoring sites are located at specific points in latitude and longitude). We selected monitoring sites for which the daily mean $PM_{2.5}$ data were available for every day of July 2019. We chose observation stations that are far enough from each other, in different U.S. states in this case, in order to compare our simulations under different atmospheric conditions, emissions, and winds.

   **Manuscript modifications:** We added descriptive location names to the panels to improve the readability of Figure 7. We also added Section S9 to the SI to expand on the choice of monitoring sites.

3. Add more information on the observations such as EPA network name, date of download, location of download, and any data processing to adjust or filter measurements.

   **Manuscript modifications:** We added Section S9 and Table S3 to provide more information on the air quality monitors and associated $PM_{2.5}$ data selected.